# Solar regulators for polar instrumentation: why night consumption matters

Michael R. Prior-Jones[1], Lisa Craw[1], Jonathan D. Hawkins[1], Elizabeth A. Bagshaw[2], Paul Carpenter[3], Thomas H. Nylen[4], Joe Pettit[5]

[1]School of Earth & Environmental Sciences, Cardiff University, Cardiff, UK
[2]School of Geographical Sciences, University of Bristol, Bristol, UK
[3]Formerly of IRIS-PASSCAL, New Mexico Tech, Socorro, New Mexico, USA
[4]Technical University of Denmark, Lyngby, Denmark
[5]EarthScope Consortium, Boulder, Colorado, USA

*Correspondence to*: Michael Prior-Jones (prior-jonesm@cardiff.ac.uk)

**Abstract.** Autonomous instruments, powered using solar panels and batteries, are a vital tool for long-term scientific observation of the polar regions. However, winter conditions, with low temperatures and prolonged lack of sunlight, make power system design for these regions challenging. Minimising winter power consumption is vital to successful operation, but power consumption data supplied by equipment manufacturers can be confusing or misleading. We measured the night

consumption (power consumption in the absence of sunlight) of 16 commercially available solar regulators and compared the results to the manufacturers' reported values. We developed a simple model to predict the maximum depth of discharge of a battery bank, for given values of regulator and instrument power consumption, solar panel size, location, and battery capacity. We use this model to suggest the minimum battery capacity required to continuously power a typical scientific installation in a polar environment, consisting of a single data logger (12mW power consumption) powered by a 12V battery bank and 20W

solar panel, for eight different types of solar regulator. Most of the tested solar regulators consumed power at or below the manufacturer's reported values, although two significantly exceeded them. For our modelled scenario, our results suggest that current consumption may be reduced by two orders of magnitude (from 23mA to 0.1mA) through careful choice of solar regulator, and the mass of the battery required for year-round operation may thus be reduced from 45kg to 1.5kg, a factor of 26x. These results demonstrate that choice of solar regulator can significantly increase the chances of successful year-round

data collection from a polar environment, eases deployment and reduces costs.

## 1. Introduction

Autonomous instruments deployed in the polar regions are often powered by solar panels and batteries (e.g. Kadokura and others, 2008; Zandomeneghi and others, 2010; Citterio, 2011; Eckstaller and others, 2022). A typical system (e.g. Zandomeneghi and others, 2010) consists of one or more solar panels, a solar regulator, a battery of calculated capacity (often 12 V lead acid), and the instrument itself. The battery may be a single unit or a battery bank consisting of multiple units connected together. A key challenge is powering the instrument during the polar winter, when there is no sunlight and the system may be exposed to temperatures below 0 °C, and so systems are designed to store energy in the battery during the summer months and then operate solely on this stored energy through the winter darkness. The power consumption of the instrument and its ancillaries versus the anticipated environmental conditions thus defines the required battery capacity. It is highly desirable to reduce this capacity, since lower-capacity batteries are physically smaller, cheaper, lighter, easier to transport, and safer to work with in the field.

Lead-acid batteries are considered here as they are commonly used for polar deployments (McGovern and Geller, 2022). Rechargeable lithium-based batteries are damaged by being recharged at temperatures below freezing (Bommier et al., 2020), and so are rarely suitable for use in a solar powered system in a cold environment. Whilst some specialist lithium rechargeable batteries are now available that use internal electronics and heaters to overcome this limitation, they are considerably more expensive than equivalent lead-acid types. Relion's 12V 100Ah low-temperature lithium battery, which is rated down to -20C, has a list price of USD $949 (Dec 2024, approx.€900), which is around three times the price of a lead-acid battery of the same capacity (Lithium Battery for Low Temperature Charging | RELiON, 2024). Whilst calculating the power consumption of the instrument is theoretically straightforward, the power consumption of the power system itself is often overlooked, particularly that of the solar regulator and any other ancillary electronics that are powered continuously (such as low-voltage disconnect circuits). Here we show how neglecting this issue negatively impacts system performance and may lead to instrument failure during the winter. In this paper, section 1.1 provides background on battery management and performance, particularly with regard to operating at low temperatures. Section 1.2 describes the function of solar regulators and introduces the two most common architectures. Section 1.3 describes what is meant by "night consumption" and the rationale for this study. Section 2.1 describes the methods for the laboratory tests on various types of solar regulator. Section 2.2 introduces our numerical

model of how a solar power system performs at polar latitudes throughout the year. Section 3 describes the results of both the lab tests (section 3.2) and the modelling work (sections 3.1 and 3.3). Section 4.1 describes limitations of the work, and section 4.2 provides design recommendations. Section 5 is the conclusion.

### 1.1. Battery management

Battery capacity depends both on load current and on temperature (Spiers, 2012). Datasheets for batteries commonly quote performance at 25°C (or 80°F = 26.6°C) and for a 20-hour discharge period. For example, a 20Ah battery is specified to produce 1A for 20 hours at 25°C. Reducing load current will increase usable capacity, as described by Peukert's equation (Ioannou et al., 2016; Peukert, W, 1897), or by curves provided by the battery manufacturers (Power-Sonic Corp., 2018; Surette Battery Company Ltd, 2020). This further rewards reducing the power consumption of instruments and associated electronics,

as it increases the available battery capacity.

Low temperatures also have a significant effect on usable battery capacity. At low temperatures, the battery electrolyte will freeze at a point dependent on the battery's state of charge. Whilst this varies between manufacturers, a fully-charged battery would typically freeze at -50°C, whereas one at 20% charge could freeze at -10°C (Labouret and Villoz, 2010; Spiers, 2012). Freezing risks damaging the battery casing and may severely hamper its performance, even after the battery has thawed. The

power system design should keep the battery sufficiently charged during the winter months to minimise the risk of freezing.

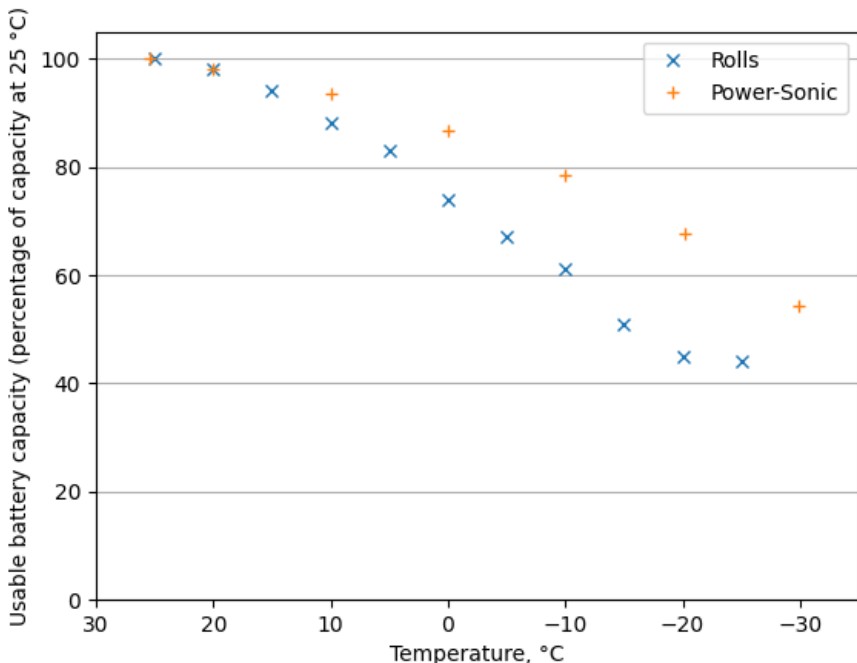

**Figure 1 – Manufacturers' recommended capacity de-rating against temperature for Rolls and Power-Sonic brand batteries discharged at 20-hour rate. Data are from technical manuals provided by both companies. Power-Sonic values were rescaled to use 25 °C as the baseline temperature. (Power-Sonic Corp., 2018; Surette Battery Company Ltd, 2020)**

Battery manufacturers provide temperature-derating curves for their batteries and Figure 1 shows them for Rolls and Power-Sonic batteries (Power-Sonic Corp., 2018; Surette Battery Company Ltd, 2020). The available battery capacity is reduced by the need to retain charge in the battery to prevent freezing. At polar temperatures of -20 °C and below, the usable battery capacity can be less than 50% of the capacity at 25 °C.

To protect the battery from being discharged too deeply and being at risk of freezing, a low-voltage disconnect (LVD) circuit

may be used. This will switch off the load if the battery voltage falls below a threshold value. It will reconnect the load when the battery recharges above a second, higher, threshold voltage. LVD circuits are built into some regulators (see Table 2) and are useful if the load's power consumption is significantly higher than that of the regulator itself. Standalone LVD units are also available, but if these are used then the standby current of the LVD must also be factored into the overall system night consumption. For example, the Blue Sea Systems m-LVD unit consumes 95 mA when the load is powered and 4mA when the

load is disconnected (m-LVD Low Voltage Disconnect - Blue Sea Systems, 2025). Galley Power's GPC series LVDs consume

3.5 mA when the load is powered and 0.26 mA when the load is disconnected (Galley Power, 2019). On some units the LVD thresholds can be programmed by the user, and the threshold values should be configured carefully based on the battery type and expected temperature conditions.

## 1.2. Solar regulators

Solar regulators are connected between the solar panels and the battery and perform two functions. Firstly, they prevent current from the battery bank from leaking back into the solar panel during the night and being wasted as heat. The majority of regulators incorporate this functionality in their internal electronics, but some (e.g SES-Flexcharge types) require an external blocking diode. Secondly, they ensure that the battery is charged in a safe and efficient manner, often by using a multi-step charging regime. A typical multi-step charging regime alters the voltage and current supplied to the battery to ensure a rapid

charge to around 80% of capacity ("bulk"), followed by a slower charge to 100% ("absorption"), then followed by maintenance at 100% ("float"). A further "equalization" cycle then occurs periodically to prevent acid stratification and sulfate buildup inside the battery (Morningstar Corp., 2022). Some manufacturers (e.g. SES-Flexcharge) implement their own proprietary charging regimes which may vary from that described here. Charging regimes may incorporate temperature compensation to improve performance, and may be customized to match the battery manufacturer's recommendations.

There are two common architectures for solar regulators: Pulse Width Modulation (PWM) and Maximum Power Point Tracking (MPPT) (Labouret and Villoz, 2010). PWM regulators are an older design and usually cheaper than MPPT regulators. In a PWM regulator, an electronic switch is pulsed on and off to regulate the output voltage. MPPT regulators (Bose et al., 1985) are a more complex design, using a switch-mode power converter to allow the solar panel and battery to operate at different voltages, improving system efficiency (Labouret and Villoz, 2010; Sunforge LLC, 2021). MPPT regulators are

slightly more expensive, and the switch-mode power converter makes them more prone to emitting electromagnetic interference (Ohba et al., 2014).

## 1.3. Night consumption

The solar regulator itself requires some power to operate, and this is described by the manufacturers as "self-consumption", "own consumption", "parasitic current", "operating consumption" or "quiescent current". The term "self-consumption" may

lead to a mistaken belief that this power is deducted from the output of the solar panel, but in nearly all regulator designs, it is consumed from the battery. Some controllers consume more current when the sun is shining, but there is almost always a baseline power consumption that is being drawn from the battery at all times. This we refer to as "night consumption". Understanding the predicted night consumption of the chosen system is critical for planning long-term instrument deployment in locations with limited solar input.

This study aims to quantify the night consumption of a range of solar regulators and confirm that the night consumption data given in the manufacturers' datasheets is correct. We also provide a modelling tool for those planning polar field deployments to calculate the required battery bank capacity that is suitable for their equipment and field site.

## 2. Methods

### 2.1. Laboratory tests on solar regulators

We tested 16 different types of solar regulator under laboratory conditions to determine if their measured night consumption agreed with the figures quoted on their datasheets. The experimental set-up consisted of connecting the regulator's battery terminals to a 12V lead-acid battery via a dual-channel multimeter (Mooshimeter, Mooshim Engineering), which simultaneously measures both the voltage and the current consumed. The solar panel input to the regulator under test was left unconnected, representing total darkness. Where an external blocking diode is recommended by the regulator manufacturer

(e.g. on SES-Flexcharge regulators) this should be connected in series between the panel and the regulator, but since we are simulating a panel in darkness with an open circuit, no current would flow through this diode in the test setup and thus it was omitted. The regulator under test was connected to the experimental set-up for 2 minutes to allow the current consumption to stabilise and then an instantaneous measurement of both current and voltage were taken. The regulator was then disconnected. The battery was not recharged between tests but the battery voltage did not drop by more than 0.1V during the tests and so the

effect of variation in battery voltage can be ignored. The current consumed in the tests was compared to that specified by the manufacturers. Tests were carried out at laboratory room temperature (approximately 20 °C) but the laboratory was not climate-controlled precisely.

The Mooshimeter multimeter in DC current mode has a 200 mΩ series resistance and the resolution for DC current measurements under 1.75 A is given as 0.4 µA with a noise floor of 10 µA (Mooshim Engineering, 2014). The smallest current measured in the experiments was 40 µA (0.04 mA), which is above the noise floor. The largest current measured while testing a regulator was 41.81mA which equates to a Thévenin equivalent resistance (Johnson, 2003) of around 300 Ω. The effect of the meter on the circuit can therefore be ignored as the meter's resistance is three orders of magnitude smaller than that of the lowest-resistance unit under test.

## 2.2. Modelling system performance

To illustrate the effect of the solar regulator's power consumption, we devised a simple spreadsheet model for the performance of a solar power system deployed in a polar environment (i.e. locations inside the Arctic/Antarctic Circles). Given the latitude and longitude of the deployment site, the model uses NOAA's Sun Calculator spreadsheet (Solar Calculator - NOAA Global Monitoring Laboratory, 2024), to determine the number of minutes of daylight for each day of the year. For each day, the model calculates the energy input from the solar panels, and then subtracts the energy consumption of both the example instrument and the solar regulator to give a net energy change per day. This net energy change then updates the energy stored in the battery from the previous day, to give a new end-of-day battery state of charge. The model calculates this for every day in a year starting from the deployment date, and finds the lowest value of the end-of-day battery state-of-charge. This is converted to depth of discharge (DoD) – 0% state-of-charge = 100% depth of discharge. This value is the model output – if the value exceeds 100%, the system will fail because the battery will run out of energy. To provide a safety margin and to give good battery longevity over multiple seasons, we recommend that this value not be allowed to exceed 60% DoD (i.e. the battery never drops below 40% state-of-charge).

We then used the model to calculate the performance of an example deployment, and varied the power consumption of the solar regulator to show how choosing different types of regulator will affect the system design. The night consumption of each regulator is taken from our experimental results, whilst datasheet values are used for the daytime consumption. For each of the

regulators modelled, we determined the minimum battery capacity (to the nearest 0.5 amp-hour) required to operate the system with the battery state of charge remaining at > 40% at the end of any single day (=60% depth of discharge (DoD).

To estimate the mass and cost of the different batteries required, we took details of 22 different types of sealed lead-acid
155  batteries for cyclic applications from the manufacturer Yuasa (the NP, NPC and REC ranges) and compared battery mass and retail price with the advertised capacity. To give the approximate mass and cost of a battery of given capacity, we interpolated between the known values from the Yuasa batteries.

The example deployment modelled is:

160  **Deployment date:** 1$^{st}$ January 2024**. Location**: 70 degrees S, 0 degrees E**.** An Antarctic location was chosen so that the example deployment year lines up with the calendar year, with deployment taking place in the middle of the summer. **Instrument current consumption = 1mA.** We took a common type of datalogger as our benchmark instrument: a Campbell CR1000X consumes around 1mA when powered from 12V and used on a 1Hz scan (Campbell Scientific, 2021). **Solar panel nominal output = 20W.** We chose a small solar panel with 20W peak output. **System voltage = 12V.** We chose a typical lead-
165  acid battery with a nominal voltage of 12V. In practice the battery voltage will vary between 11.5V and 14.9V depending on circumstances (Morningstar Corp., 2022).

The model uses a number of following parameters and assumptions, which are described below. We aimed for conservative estimates for these parameters so that the results tend towards an overestimate of the battery capacity required, giving some margin for unforeseen effects.

**Solar panel yield = 10%**. We assume that the solar panel produces an average yield of 10% of its rated peak output during daylight hours. For a 20W panel this means that it will produce an average of 2W during the hours of daylight. This value is a conservative estimate, using an installation at Showa station at 70 degrees South in Antarctica as a benchmark (Frimannslund et al., 2021; Tin et al., 2010). The Showa system has a specific yield (total energy produced per year per kilowatt-peak of installed panel capacity) of 800kWh/kWp/year, which was converted to average yield (joules per second of daylight per watt-peak of installed panel capacity), using the NOAA Sun Calculator (Solar Calculator - NOAA Global Monitoring Laboratory, 2024) to determine the total annual daylight at Showa, resulting in an average yield of 17.9%. To account for the large-scale installation at Showa being highly optimized for yield, and to acknowledge that most small-scale installations will be lower-yielding as a result of non-optimal installation, local weather conditions, cloud cover and panel degradation, we chose a figure of 10% average yield for the model.

**Starting battery state-of-charge = 50%**. Whilst lead-acid batteries are charged before they leave the factory, they experience a self-discharge of around 1% per month (Surette Battery Company Ltd, 2020). Whilst ideally batteries would be deployed to the field fully charged, this is not always practical, and so we have looked for a conservative estimate of a likely starting state-of-charge. We chose a value of 50%, which represents a relatively low state-of-charge, covering the situation where the battery has been stored without charging before deployment..

**Low-temperature battery capacity reduction factor = 50%.** Battery manufacturers recommend a variety of de-rate factors for low temperatures depending on the exact battery chemistry and the power consumption of the load (Power-Sonic Corp., 2018; Surette Battery Company Ltd, 2020). However, 50% is a typical value for temperatures of -20ºC. Therefore, a battery sold as 100Ah at 25ºC is considered by the model to have 50Ah of capacity in a polar environment (based on operating at -20ºC).

**Maximum depth of discharge = 60%.** There is a tradeoff between depth of discharge and cycle life – the number of times that the battery is fully charged and then discharged below a threshold level - often 60% DoD - in the battery's lifetime. For off-grid solar power installations outside the polar regions the battery may experience one cycle per day, so battery manufacturers recommend limiting depth of discharge to 50% to maximize cycle life (Surette Battery Company Ltd, 2020).

However, for a polar installation the battery experiences one deep cycle per year, so maximizing cycle life is much less important, as even at higher DoDs the cycle life is measured in thousands of cycles. We chose 60% DoD as our threshold value.

**Charge efficiency = 85%.** We model the battery using a simple coulomb-count model (Ng et al., 2009). The charge efficiency parameter models how much of the energy delivered to the battery during charging is output during discharge. The figure of

85% on average is quoted by (Stevens and Corey, 1996). We do not model the variation of charge efficiency based on battery state of charge, but this could be included in future work.

## 3. Results

### 3.1. Solar regulator laboratory tests

The datasheet and measured night consumption values of the regulators are presented in Figure 23 and Table 1. A comparison between datasheet and measured night consumption values for each regulator is shown in Figure 3.

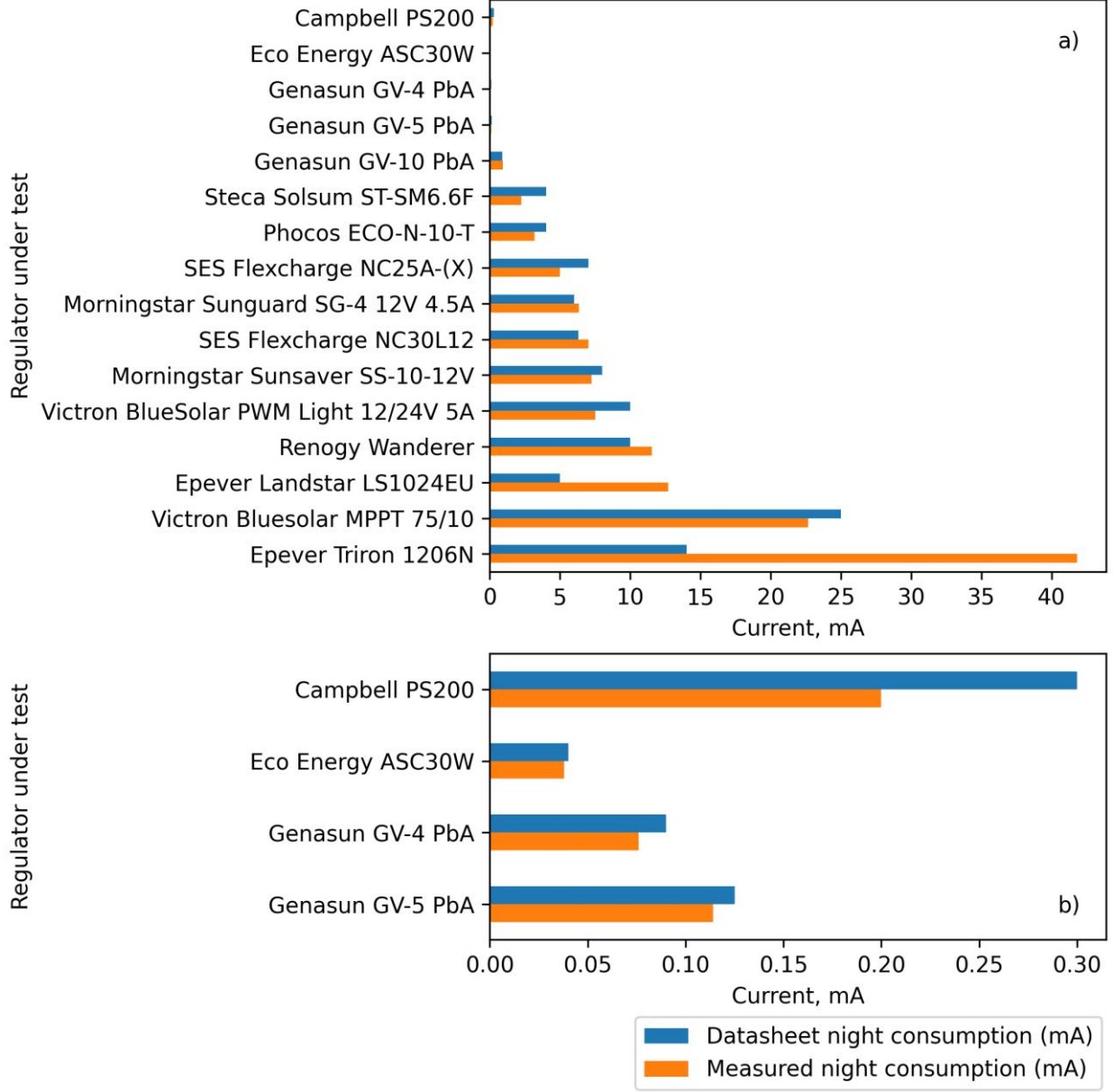

**Figure 2 – night current consumption of regulators tested. Panel a) shows all regulators tested; panel b) is a zoomed-in version to show the relative current consumption of the four regulators with the lowest night consumption**

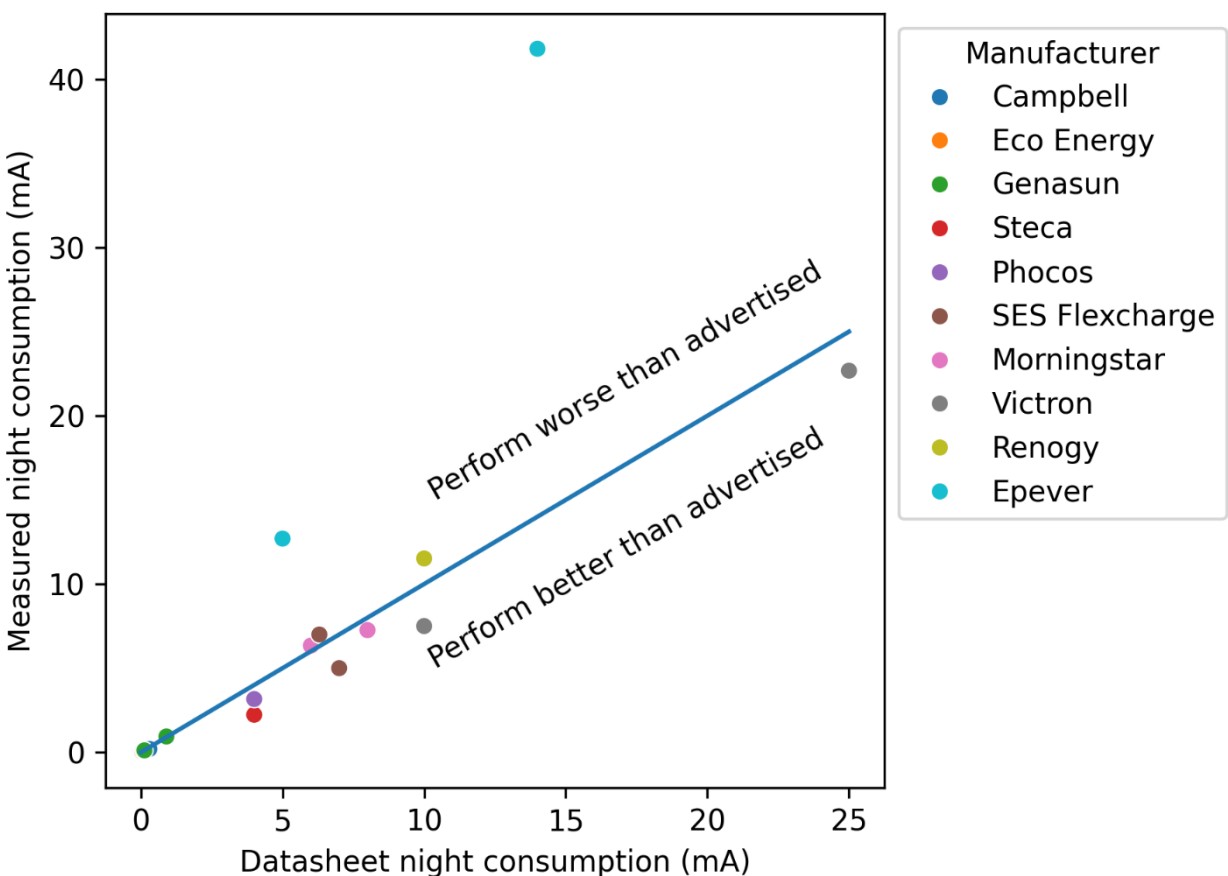

**Figure 3 - scatter plot showing measured night consumption vs datasheet values. The blue line shows the 1:1 relationship, so points plotted below the line indicate regulators where the night consumption was below that stated on the datasheet, whereas those above the line have night consumptions that are in excess of the datasheet value.**


**Table 1: Manufacturer's specifications and approximate 2023 purchase prices for all regulators tested in this study, alongside measured night consumption when attached to a 12.5V battery.**

| | Information from manufacturers' datasheets | | | | | | | Measured data | |
|---|---|---|---|---|---|---|---|---|---|
| Manufacturer | Type number | Architecture | Temperature compensation? | LVD built-in? | Approx price (EUR) | Rated solar panel current (A) | Datasheet night consumption (mA) | Measured night consumption (mA) | Measured night consumption as a percentage of datasheet value |
| Campbell | PS200 | PWM | Yes | No | 460 | 3.6 | 0.3 | 0.2 | 67% |
| Eco Energy | ASC30W | PWM | Yes | No | 72 | 2.5 | 0.04 | 0.04 | 100% |
| Genasun | GV-4 PbA | MPPT | Yes | No | 77 | 4 | 0.09 | 0.08 | 89% |
| Genasun | GV-5 PbA | MPPT | Yes | Yes | 90 | 5 | 0.125 | 0.11 | 88% |
| Genasun | GV-10 PbA | MPPT | Yes | No | 115 | 10.5 | 0.9 | 0.94 | 104% |
| Steca | Solsum ST-SM6.6F | PWM | Yes | Yes | 31 | 6 | 4 | 2.23 | 56% |
| Phocos | ECO-N-10-T | PWM | Yes | Yes | 66 | 10 | 4 | 3.16 | 79% |
| SES Flexcharge | NC25A-(X) | PWM | No | No | 137 | 25 | 7 | 5.00 | 71% |
| Morningstar | Sunguard SG-4 12V 4.5A | PWM | Yes | No | 50 | 4.5 | 6 | 6.34 | 106% |
| SES Flexcharge | NC30L12 | PWM | Optional | Yes | 215 | 30 | 6.3 | 7.00 | 111% |
| Morningstar | Sunsaver SS-10-12V | PWM | Yes | No, but sister type SS-10L-12V does | 80 | 10 | 8 | 7.26 | 91% |
| Victron | BlueSolar PWM | PWM | No | Yes | 30 | 5 | 10 | 7.50 | 75% |

| | Light 12/24V 5A | | | | | | | | |
|---|---|---|---|---|---|---|---|---|---|
| Renogy | Wanderer | PWM | Yes | Yes | 23 | 10 | <10 | 11.52 | 115% |
| Epever | Landstar LS1024EU | PWM | Yes | Yes | 23 | 10 | 5 | 12.70 | 254% |
| Victron | Bluesolar MPPT 75/10 | MPPT | Yes | Yes | 66 | 10 | 25 | 22.68 | 91% |
| Epever | Triron 1206N | MPPT | Yes | Yes | 96 | 10 | 14 | 41.81 | 299% |

In Table 1, the datasheet values of night consumption required some calculations and assumptions. SES Flexcharge and Epever specify power consumptions for different parts of the circuit which were totalled to give night consumption. For the Eco Energy ASC30W, the datasheet night consumption is stated as "0.0mA". We have taken the value as 0.04mA, which is the largest number to two decimal places that rounds down to 0.0mA at one decimal place.

The results in Table 1 and Fig. 3 show that the night consumption was below the manufacturer's estimates for the majority of regulators but there were six units with consumption in excess of the datasheet values.

Some types performed better than expected, notably the Steca Solsum ST-SM6.6F, where manufacturer night consumption was 4 mA but measured consumption was 2.3 mA. Campbell PS200 (0.3 vs 0.2 mA), Phocos (4 vs. 3.16 mA), SES Flexcharge (7 vs. 6.3 mA) and Victron Bluesolar (PWM 10 vs. 7.5 mA, MPPT 25 vs. 22.68 mA) also performed better than advertised. However, the two Epever regulators both had measured performance that was far worse than their datasheet claimed (Landstar 5 vs. 12.70 mA and Triron 14 vs. 41.81 mA). The Renogy performed slightly worse than specified (<10mA vs 11.5mA).

It is worth comparing these figures with the 1mA load current for our example Campbell datalogger: for the Eco Energy ASC30W, the current consumption of the regulator is 4% of the load current. By contrast, the Victron Bluesolar MPPT75/10 has a night current of almost 23x that of the load – which would result in 96% of the system's power consumption going into running the solar regulator.

Historically, MPPT regulators had higher night consumption than PWM regulators due to the additional circuit complexity (Labouret and Villoz, 2010), but our results show that the newest MPPT designs have lower night consumptions than many

PWM types.. Taking regulators with 10A solar input rating as an example, the MPPT Genasun GV-10 PbA has a <1 mA night consumption, which is lower than the PWM Phocos (3.16 mA) and Morningstar SS-10-12V (7.26 mA).

### 3.2. Relationship between battery capacity, mass and price.

Figure 4Figure 1a shows the mass of Yuasa lead-acid batteries when compared with battery capacity. These values are from the Yuasa datasheets in December 2024.  Fig. 4b shows the relationship between battery capacity and December 2024 purchase

prices. The trendlines show that both vary linearly with battery capacity, and the equations on the graph can therefore be used to estimate the price and mass of a battery of a given capacity.

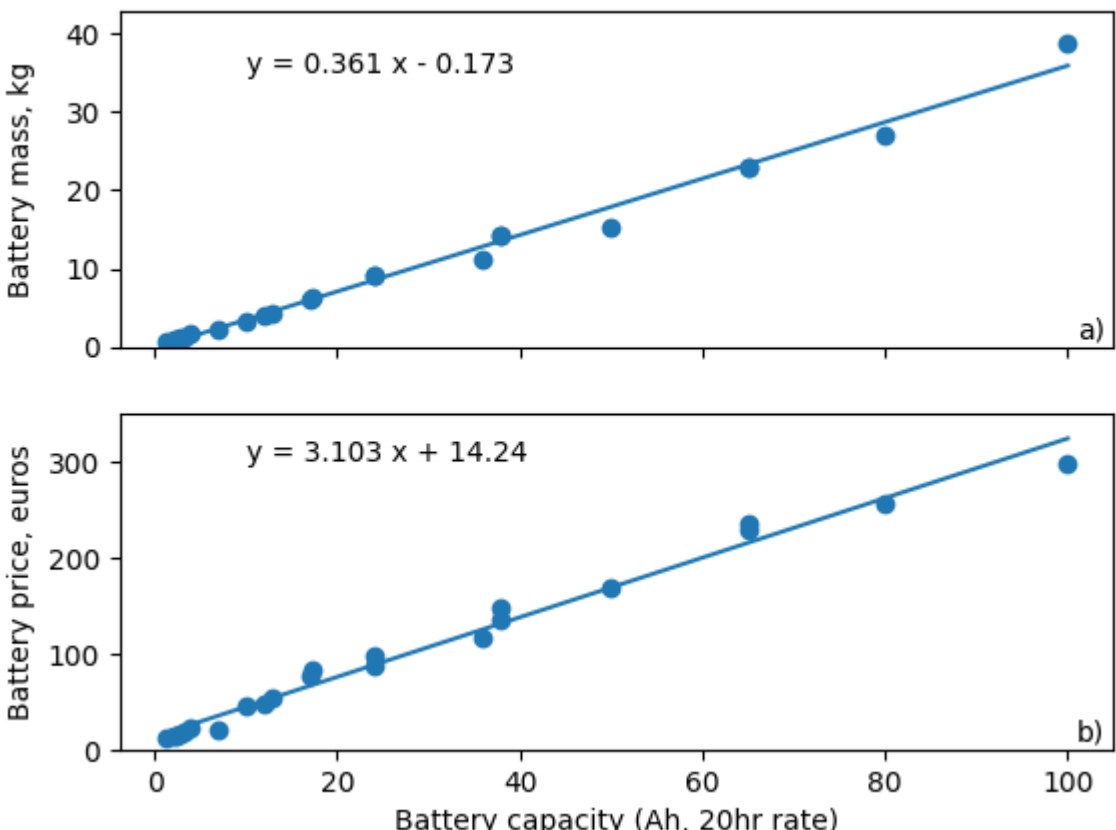

**Figure 4 - linear relationships between battery capacity, mass and price for Yuasa lead-acid batteries. Batteries chosen were from the NP, NPC and REC ranges. Mass values are from the Yuasa datasheets. Prices were obtained in December 2024 from a UK supplier and converted from British pounds to Euro at an exchange rate of 1 GBP = 1.2 EUR.**

### 3.3. Model results

**Table 2: Modelled battery capacities and masses required to maintain >40% battery capacity (<60% DOD) for one year with a selection of solar regulators tested in this study. Battery masses and prices are estimated using Yuasa battery data in Fig. 4. Calculations are based on a scenario where an instrument consuming 1mA is deployed on the 1$^{st}$ of January at 70° S with a 20W solar panel and 12.5V battery.**


| Manufacturer | Type | Minimum battery capacity (25C, 20-hr) required, as calculated by model (Ah) | Estimated battery mass (kg) | Estimated battery price (Euro) | Additional battery mass required due to solar regulator (kg) |
|---|---|---|---|---|---|
| Perfect regulator | | 4.5 | 1.5 | 28 | 0 |
| Genasun | GV-4 | 5 | 1.7 | 30 | 0.2 |
| Genasun | GV-5 | 5 | 1.7 | 30 | 0.2 |
| Eco Energy | ASC30W | 5 | 1.7 | 30 | 0.2 |
| Campbell | PS200 | 6 | 2 | 33 | 0.5 |
| Steca | Solsum ST-SM6.6F | 14.5 | 5.1 | 59 | 3.6 |
| Phocos | ECO-N-10-T | 21 | 7.4 | 80 | 5.9 |
| Morningstar | Sunguard SG-4 | 33.5 | 11.9 | 119 | 10.4 |
| Victron | Bluesolar MPPT 75/10 | 125 | 45 | 402 | 43.5 |

Table 2 shows the different battery capacities required to support the power consumption of several of the regulators tested under these hypothetical deployment conditions (see section 2.2). These results are the minimum battery capacity required to operate the instrument year-round for a given solar regulator. A theoretical "perfect" regulator is also shown for comparison.

Battery price and mass estimates are calculated based on the battery capacity, using the equations from Figure 4Figure 1. In the worst-case example modelled here, choosing the Victron Bluesolar MPPT 75/10 would require a battery capacity of at least 125Ah. A battery of this capacity is over 45kg and costs more than €400. Choosing an alternative regulator with lower power consumption allows for the use of a smaller battery: the inexpensive Steca Solsum ST-SM6.6F needs only a 14.5Ah battery, with a mass of around 5.1kg. The Genasun GV-4, Genasun GV-5 and Eco Energy ASC30W offer the best overall

performance, allowing the use of a 5Ah battery, with a mass of <2kg. Choosing one of these best-performing regulators results in a 26x reduction in battery mass and a 13x reduction in battery price when compared with the Victron MPPT 75/10.

## 4. Discussion

### 4.1. Limitations

This study did not aim to simulate the full polar environment, and so we have not tested the effects of low temperature on the

night consumption of the regulators. Further practical limitations include the limited range of solar regulators tested, the fact that the tests were only conducted for a short period of time, and the use of an open circuit instead of a solar panel in darkness. It was not practical for us to measure the daytime consumption of the regulators and verify this against the datasheet values – this would have required the use of either a calibrated illumination chamber and solar panels, or an electronic solar panel simulator, neither of which we had access to.

The spreadsheet model does not consider the effect of snow accumulation covering solar panels, nor the effect of mountains or other topography shading the solar panels when the sun is at low angles. It also neglects the effect of state-of-charge on charge efficiency and the variation in usable battery capacity with load current.

### 4.2. Design recommendations for polar field deployments

Beyond careful choice of solar regulator and modelling of system performance, we also recommend:

• Minimising the power consumption of the instrument itself – such as by powering down sensors between measurements.

• Using a solar regulator with temperature compensation.

• Using a low-voltage disconnect circuit (LVD) to protect the battery from freezing.

- Choosing LVD thresholds carefully – having a relatively high voltage for the "reconnect load" voltage helps ensure that the battery is well-recovered before it delivers power to the load.

- Using a well-insulated enclosure (Clauer et al., 2014; Musko et al., 2009) for the batteries, especially if the electronics are included in the battery box, as the insulation will help retain heat from the electronics.

- Increasing the battery capacity by a "factor of safety". For a system with low power consumption, it may be practical to increase the battery capacity as predicted by our model by 50 or 100% to provide additional resilience. For more power-hungry systems a smaller factor may be used. Consider the number of "days of autonomy" (i.e. how long can the system run on battery power alone) and increase battery capacity based on data about local conditions.

- Increasing the modelled solar panel capacity by a "factor of safety". Solar panel performance can be severely degraded by snow or ice frozen onto the panels, or by shadowing from local topography. Using larger panels (or more, smaller panels) can help ensure that the system produces the necessary power during the crucial spring period when the battery levels are at their lowest.

- Mounting the solar panels vertically (to minimise snow retention) and facing toward the Equator. If there is local topography blocking the direct sightline, consider multiple panels facing in different directions (e.g. one facing east and one west).

- Considering that large solar panels can have significant wind loading and need to be appropriately supported. Multiple smaller panels may be easier to manage than one large one.

- Planning for local snow accumulation when choosing the mounting for your panels – they need to be mounted high enough to be out of the snow in the spring.

- Testing the performance of your entire system before deployment, ideally in a cold chamber to ensure that everything performs as expected at polar winter temperatures.

## 5. Conclusions

When designing a solar-powered autonomous instrument for use in the polar regions, it is vital to consider the power consumption of the whole system, not just the instrument itself. We provide a workflow for assessing the night-time power

consumption of solar regulators and test 16 of them in this study. These laboratory tests demonstrate that the power consumption of solar regulators varies significantly, and that values in manufacturers' datasheets are not always trustworthy.

We also provide a calculation tool for polar field scientists to calculate the required battery capacity based on solar regulator specifications, to assist in choosing a solar regulator that is fit for purpose. In our own calculation example this resulted in a 26x reduction in required battery weight and 13x reduction in battery cost.

Careful choice of solar regulator (and any other ancillary electronics) can prevent the equipment from running out of power and damaging the batteries during the polar winter. It can significantly reduce the size and mass of the battery required for a

successful deployment, which reduces both purchasing and logistics costs. Power calculations should therefore include the 'night consumption' of the regulator and other ancillaries when designing systems for autonomous operation in the polar regions, particularly over winter. Careful power system design can reduce the cost and complexity of purchasing, deploying and operating instruments and increase the chances of successful, year-round data collection.

**Data availability**

Data and Python code for the plots in Figs. 1-4 is available via Zenodo https://doi.org/10.5281/zenodo.17457595 or at https://github.com/CHILCardiff/solaregulators-plots/releases/tag/post-review-v2. Any future updated version will be available at https://github.com/CHILCardiff/solaregulators-plots.

**Code availability**

The spreadsheet used for the modelling in this paper is provided in Microsoft Excel and OpenDocument Spreadsheet

formats. Instructions for its use are provided within the spreadsheet and in the README.md file. The released version at the time of publication may be obtained from Zenodo https://doi.org/10.5281/zenodo.15115132 or at https://github.com/CHILCardiff/solarregulators-model/releases/tag/preprint-v1. Any future updated version will be available at https://github.com/CHILCardiff/solarregulators-model.

## Author contribution

MPJ conceived the study. MPJ, LC, JH, PC, TN and JP made measurements of solar regulator performance. MPJ and JH developed the spreadsheet tool. All authors contributed to the manuscript. This is Cardiff EARTH CRediT Contribution 44.

## Competing interests

The authors declare that they have no conflict of interest.

## Acknowledgments

Thanks to members of the Cryolist mailing list for suggesting regulators to test, and to Midsummer Energy, Samuel H Doyle, and Campbell Scientific for the loan of several types of solar regulator. We thank Rolf Hut and one anonymous reviewer for their helpful suggestions and comments. Work for this study was funded by EPSRC grant EP/R03530X/1 (to EB) and UKRI Future Leaders Fellowship MR/V022237/1 (to MPJ).

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
