# Peer review of "Solar regulators for polar instrumentation: why night consumption matters"

_EGUsphere, 2025_

## Referee Comment (RC2)

**Review of Solar Power system for polar instrumentation: why night consumption matters**

Article by M. Priot-Jones et. al. Review by Rolf Hut.

The authors set out to test whether solar regulators, the devices in between solar panels, batteries and loads that manage charging of batteries, perform as advertised in their datasheets with respect to the nighttime consumption of these devices. They furthermore provide a calculation tool for calculating optimal battery size for a given combination of solar panel, solar regulator and instrumentation load. Their focus is solidly on using solar panels and instrumentation in an arctic setting, dealing with both very low temperatures and the long arctic night.

I think this work is valuable to the community and of interest to the readership of GI. I believe (knowing of) this work will help other scientist who set out for (arctic) field campaigns plan better and avoid either high costs or system failure by applying the lessons from this work. In that regard I philosophically disagree with reviewer #1, who stated

> I think it is too simplistic to assume one would pick a solar regulator, put together a sensor-logger system and place it in the arctic. What people will usually do is – as suggested in the manuscript – a dry run and logging of the energy consumption. So, there will be extensive information on the power consumption already available.

This statement by reviewer #1 in my opinion provides too sunny a picture of the daily work of many scientists. It is my hypothesis that many experiments brought to the field in general and the arctic in particular fail because not everyone is aware of these issues with batteries and solar generators. Those failed experiments hardly make the scientific literature because of systemic problems with publishing negative results. Scientists that have (access to) a lab equipped with tools and staffed by electrical engineering technicians to execute and help with a dry run might have the option to do so, making their chances of successful field campaigns higher. Even they are helped by the work of Prior-Jones et. al. in the sense that they can make the decision on which device to buy for these tests informed.

The main take-away from the work is that datasheets should not be blindly trusted. I strongly believe that as a community, when a scientist has tested a device and either confirms the datasheets, or has contractionary findings, this is valuable information for the entire community, avoiding double work for other scientists. Sharing these results therefore have my strong support.

I do, however, have suggestions to improve on the clarity and readability of how the work is presented. I therefore recommend minor changes be made to the manuscript before publishing.

**clarity**

The authors employ a loose style of writing suited for easy reading, but this does occasionally lead to the function of different parts of the document being mixed or unclear. For example, starting on line 61 the authors provide a recommendation in the introduction. This feels out of place as the introduction is meant for "setting the scene".
For further clarity I recommend:
1. Adding a small reading guide after line 47: "in the rest of this document, in paragraph 1.1 we will review the current state of the art in battery management, in 1.2 on solar regulators …. Etc.
2. Move "but we show …" on line 100 to the results and out of the introduction.
3. Focus the summary of the goal at the end of section one: aim to check if datasheets reported values are correct, plus, we provide a modelling tool for arctic field workers to calculate required battery size for a given setup
4. Start a new paragraph after line 124 to signal the break between description of the model and the application of the model with a given set of assumptions
5. Change paragraph 4 into "discussion" and split into 4.1: discussion on this study (or a better title) and 4.2 design recommendations for arctic fieldwork. In 4.1 add a few lines on the limitations of this study: limited selection of solar generators, short time measurement of night time consumption as represented of night time consumption throughout arctic night, temperature effects of arctic night on night time consumption not modelled in the lab, etc.
6. In the conclusion, re-emphasize the added value of this work to the community: add a sentence to the like of: "we provide a workflow for assessing the night time consumption of solar regulators and test X of them in this work, showing that datasheets are not typically to be trusted regarding night time consumption. Furthermore we provide a calculation tool for arctic field scientist to calculate required battery size based on solar regulator specs and therefore aid in choosing the right solar regulator fit for purpose. In our own calculation example this resulted in a 26x reduction in required battery weight and 13x reduction in battery cost."

**Rigor**

At points I missed information to fully understand the work done and presented. In particular:
1. What is the impedance of the multimeter used? Does this effect the measurements in any way (for devices with very small nighttime consumption?)
2. The authors state that some regulators have diodes that prevent back currents through the solar panels during nighttime, but others do not. How does not connecting a solar panel during the tests relate to this?

3. For the results in figures 1 and 2, please indicate if these are modelled, measured values, or values reported by manufacturers. (I believe this is the latter, right?) Please provide a timestamp for these values as well both in the main text as well as in the figure caption (I believe these are 2023 values?)
4. In table 1, make a clear visual divide between reported values and measured values. I also recommend adding a column that provides the ratio (or a % increase / decrease) of the measured value versus the datasheet value.
5. In table 2 I find the double use of the word "model" confusing, both for the numerical model presented in this work and for specific solar regulators.
6. Also in table 2: please add a line for "perfect regulator" where you assume no nighttime consumption at all. I also suggest (but leave to the authors to decide) to add two columns "additional battery weight required because of solar regulator" and "additional weight because of more nighttime consumption measured than solar regulator reports in datasheet". This last one is mainly for the press release or popular blog if the authors decide to write those ;-).
7. Please link to version releases of the github repository, not to the general repository (ie. to https://github.com/CHILCardiff/solarregulators-model/releases/tag/preprint-v1) If the authors intend to update the repository in the future with more measurements or more regulator-models, this link points to the version used in this paper and they can add a sentence saying that the general version might be updated later.

---

## Author Response (AR1)

On behalf of my co-authors, please accept my thanks to the reviewers and handling editor for their time and effort in reviewing this paper. Please also accept my apologies for the delay in providing the updated manuscript – the preparations for my fieldwork this year took up a lot more of my time than I had planned for.

Here are my point-by-point responses to the reviewers' comments – my responses are in dark red text.

**Reviewer 1**

In the manuscript I reviewed, the authors compare the power consumption of sixteen commercially available solar regulator devices to evaluate their impact on the energy design of instrumentations in polar regions, where the long polar night poses the challenge of long time dark and cold conditions. The study design is simple but adequate for the narrow, envisioned goal. The overall study is relevant and timely. It might be a proper contribution to the journal and of help especially for people who wish to put together an autonomous system without extensive testing, benchmarking and optimising, although it is clearly not a rocket science paper.

I agree – this is not intended to be "rocket science" but it is still something that it is useful to the polar science community.

That said, I shall note that the overall depth of the study – apart from the laudable goal of summarising the specifics of commercially available solar regulators and maybe putting together some specifics of lead battery models from one manufacturer – is rather poor. Many of the figures are redundant, poor in content and sometimes convoluted in their way of delivering information. Hence, in principal this manuscript is eligible to become published, if the editors decide that sufficient scientific depth is given and some improvements on the presentation are implemented.

We have taken these comments on board and redesigned the figures.

In very general terms, I think it is too simplistic to assume one would pick a solar regulator, put together a sensor-logger system and place it in the arctic. What people will usually do is – as suggested in the manuscript – a dry run and logging of the energy consumption. So, there will be extensive information on the power consumption already available. Nevertheless, it may be helpful to have an overview of what is available on the market of solar regulators. In any way, the authors should down tone their way of framing the study a bit, to get closer to a scientist's daily work reality.

We disagree on this, as previously communicated. Part of the inspiration for this paper was seeing how scientists based in university departments, especially those without access to electronic engineering technician support, make poorly-informed choices when it comes to the engineering of their equipment. Reviewer 2 supports us on this point.

The text sometimes struggles to provide quantitative information but instead stays vague and descriptive. I list some of these issues in the details below. I strongly encourage the authors to check their text and add quantitative information where possible as this would increase the weight of arguments and help readers to retrieve key information.

The text has been revised – please see the notes below.

I find the term "night consumption" not really much more helpful than the other terms listed by the authors. Ideally, we are interested in the combined effect of "night and day" consumption. It is fair to stick to "night" conditions only for the specific case of polar deployments over winter time, but a more realistic case will be "day" deployments with fuzzy to cloud cover reduced sunlight, too. Hence, if I were to buy solar regulators, I would be keen on knowing the power consumption also during "day" conditions when incoming solar power – however weak it may be – needs to be managed, too. It would be good to at least mention this aspect and justify why, in the narrow context of polar nights, it is not considered by the study.

This would have made the study considerably more complex, as to study the daytime consumption properly would have required either the use of a real solar panel and a calibrated illumination chamber, or a "solar panel simulator" (essentially a programmable power supply configured to emulate the voltage-current characteristics of a solar panel), neither of which we have access to. We have added some commentary on this to the new "limitations" section.

In the title, it would help to add "solar regulators" to be specific and give the implication that the study indeed focusses on that one item rather than on power consumption of the entire system.

Changed as requested.

l. 13, remove "uniquely", no need to emphasise

Changed as requested.

l. 22, "factor of 26x" reads cumbersome, replace by something like "reduction to xx %". In addition, converting power consumption to battery weight reduction is not wrong but also it dilutes the crispness of the message. I suggest to report on the percent power reduction and then, in a next sentence, report on several of the consequences of that, which includes reduction in battery capacity, costs and weight.

This has now been reworded.

l. 23, "good choice", remove "good"

Changed as requested.

l. 42, "3x" replace by "three times"

Changed as requested.

l. 51, it is wrong that the load changes the capacity of the battery. That will always be the same. The sentence should be changed to say that increasing the load will reduce the record span, the time over which a battery of a given capacity can supply power to a system.

Respectfully, the reviewer is wrong on this point. Usable battery capacity is a function of load, as described by Peukert's equation (Ioannou et al., 2016; Peukert, W, 1897). With a smaller load, more energy can be extracted from the battery. This has been clarified in the text.

l. 55, it would be good to provide some graph here instead of arbitrary listing some charge numbers for some temperatures. In line 59, there are references to such graphs.

I really would like to see the relationship between temperature and charge as this relationship is non-linear and would help readers to better understand the problem of cold conditions and energy design.

It was not possible to reproduce the graph from the cited textbook without payment of a copyright fee, but we have instead provided a graph based on data from two battery manufacturers' datasheets which we believe conveys the intended message about the non-linearity and indeed how much the temperature effect varies between manufacturers.

l. 65, please give an overview of the range of power consumption of such LVD units, so that readers can get a feeling for the burden arising from them.

Provided as requested.

l. 79-92, this section about solar regulator architecture is overly long and clearly off scope. I encourage the authors to shorten it by at least 50 %. All what is needed is a basic understanding of the two designs (PWM and MPPT) and the consequences.

Shortened as requested.

l. 110, please add that (I assume this was done in the test) the battery was always recharged before each test sequence. Add information on ambient temperature. Add information on Multimeter specs. Add information on the current measurement interval. Check, there are maybe further test specifications that need to be delivered to make the study reproducible.

We have clarified the test procedure and provided additional information around the specifications of the multimeter.

l. 135, personally I am not a big fan of bullet points as they disrupt (intentionally) the flow of text and semantics. Make sure this is what you want to do with this style element in a "flow text piece of scientific communication".

The bullet points have been removed.

l. 148, there is some copy-paste fragment in this line. Please revise.

Revised as requested.

l. 156, not really meaningful in my opinion. Usually I would charge a battery fully full before dumping it in a hostile environment for many months. You may want to justify the assumed 50 % discharge level.

Reworded to make the justification clearer.

l. 179-180, two lines of text are a fair bit from enough to make a paragraph or even a chapter. Revise structure to be more meaningful.

Fig.1, I would combine fig.1 and fig. 2 into a two-panel figure and discuss these issues together. This might also solve the above comment of mine.

We combined Figs 1 and 2 and reworded the text, and moved this section to below the results from the lab tests.

All figures: remove the title from the plots as the figure caption is about to give that information.

Changed as requested.

Fig. 3-5, these are arbitrary split visualisations of the same type and content. The data should be moved to a single figure, perhaps separated by thin vertical lines between the suggested classes. Although, I do not see a need for these classes (4.5, 7 and 9 A). If these are to be kept they need to be justified and explained.

Moved to a single two-panel figure, with the lower panel providing a zoomed view of the four regulators with the lowest power consumption.

I rather would like to see more figures of other aspects on the data. These could include: 1) A_rated versus A_measured as scatter plots (A classes as different symbols, and a 1:1 line) to focus explicitly on the difference of the two metrics rather than indirectly via bar plots; 2) price versus percent deviation of measured from rated Ampere values, 3) metrics separated by technology (PWM and MPPT), or further explicit tests and illustrations.

We have included the scatter plot showing the measured vs datasheet night consumption, with the points coloured by manufacturer. We considered the other analyses but do not feel they add further value.

Fig. 4, X-axis label is not meaningful, revise.

This figure has been removed in favour of a new combined figure.

Table 2 should also provide an estimate of the record length rather than the "minimum battery size required".

This is a misunderstanding of the methodology on the reviewers' part and so we have provided additional clarification in the text.

l. 235, give information on which devices have temperature compensation as column in table 1

Provided as requested

l. 236, define "LVD" (again, since it is a long time since its last usage)

Changed as requested

l. 239, well but better insulation also prevents sunlight from warming up the casing, and I doubt that any insulation is good enough to prevent a battery from cooling under arctic conditions for many months. So, this argument does not make sense to me in the specific scope of the study.

We disagree and have provided two references showing that this is done in Antarctic instrumentation.

l. 241-244, 256-257, these topics are really trivial information, I would suggest to remove them. l. 245-246, 249-255, these topics are way off scope. Consider removing.

We disagree – this is hard-won practical advice that many scientists and engineers will find of value when planning their future fieldwork. However, if the editor feels that this whole section is not appropriate to the journal, we will remove it.

**Reviewer 2**

I'm grateful to Dr Hut for his comments, particularly those on the value of the study in the first section of his review. Responding to his specific points on the writing (again, my comments in red):

The authors employ a loose style of writing suited for easy reading, but this does occasionally lead to the function of different parts of the document being mixed or unclear. For example, starting on line 61 the authors provide a recommendation in the introduction. This feels out of place as the introduction is meant for "setting the scene".

This has been reworded.

For further clarity I recommend:

1. Adding a small reading guide after line 47: "in the rest of this document, in paragraph 1.1 we will review the current state of the art in battery management, in 1.2 on solar regulators …. Etc.

Added as requested

2. Move "but we show …" on line 100 to the results and out of the introduction.

Changed as requested            .

3. Focus the summary of the goal at the end of section one: aim to check if datasheets reported values are correct, plus, we provide a modelling tool for arctic field workers to calculate required battery size for a given setup

Changed as requested

4. Start a new paragraph after line 124 to signal the break between description of the model and the application of the model with a given set of assumptions

Changed as requested

5. Change paragraph 4 into "discussion" and split into 4.1: discussion on this study (or a better title) and 4.2 design recommendations for arctic fieldwork. In 4.1 add a few lines on the limitations of this study: limited selection of solar generators, short time measurement of night time consumption as represented of night time consumption throughout arctic night, temperature effects of arctic night on night time consumption not modelled in the lab, etc.

Changed as requested.

6. In the conclusion, re-emphasize the added value of this work to the community: add a sentence to the like of: "we provide a workflow for assessing the night time consumption of solar regulators and test X of them in this work, showing that datasheets are not typically to be trusted regarding night time consumption. Furthermore we provide a calculation tool for arctic field scientist to calculate required battery size based on solar regulator specs and therefore aid in choosing the right solar regulator fit for purpose. In our own calculation example this resulted in a 26x reduction in required battery weight and 13x reduction in battery cost."

Reworded taking this suggestion on board.

Rigor

At points I missed information to fully understand the work done and presented. In particular:

1. What is the impedance of the multimeter used? Does this effect the measurements in any way (for devices with very small nighttime consumption?)

We have added additional information on this.

2. The authors state that some regulators have diodes that prevent back currents through the solar panels during nighttime, but others do not. How does not connecting a solar panel during the tests relate to this?

A sentence has been added to clarify this.

3. For the results in figures 1 and 2, please indicate if these are modelled, measured values, or values reported by manufacturers. (I believe this is the latter, right?) Please provide a timestamp for these values as well both in the main text as well as in the figure caption (I believe these are 2023 values?)

Changed as requested.

4. In table 1, make a clear visual divide between reported values and measured values. I also recommend adding a column that provides the ratio (or a % increase / decrease) of the measured value versus the datasheet value.

Changed as requested.

5. In table 2 I find the double use of the word "model" confusing, both for the numerical model presented in this work and for specific solar regulators.

We have changed the terminology throughout the paper, to use the word "type" when referring to the specific regulator and use the word "model" only to refer to the numerical model. We also used the word "architecture" rather than "type" in the table to refer to whether a regulator as PWM or MPPT.

6. Also in table 2: please add a line for "perfect regulator" where you assume no nighttime consumption at all. I also suggest (but leave to the authors to decide) to add two columns "additional battery weight required because of solar regulator" and "additional weight because of more nighttime consumption measured than solar regulator reports in datasheet". This last one is mainly for the press release or popular blog if the authors decide to write those ;-).

We have added the "perfect regulator" row and the "additional weight because of solar regulator" column.

7. Please link to version releases of the github repository, not to the general repository (ie. to https://github.com/CHILCardiff/solarregulators-model/releases/tag/preprint-v1) If the authors intend to update the repository in the future with more measurements or more regulator-models, this link points to the version used in this paper and they can add a sentence saying that the general version might be updated later

This is a good idea – thank you. Updated as requested.